# Testing the Biocontrol Ability of a *Trichoderma*-Streptomycetes Consortium against *Pyrrhoderma noxium* (Corner) L.W. Zhou and Y.C. Dai in Soil

**DOI:** 10.3390/jof9010067

**Published:** 2022-12-31

**Authors:** Harrchun Panchalingam, Nina Ashfield-Crook, Vatsal Naik, Richard Frenken, Keith Foster, Russell Tomlin, Alison Shapcott, D. İpek Kurtböke

**Affiliations:** 1School of Science, Technology and Engineering, The University of the Sunshine Coast, 90 Sippy Downs Dr, Sippy Downs, QLD 4556, Australia; 2Frenkenscapes Pty Ltd., Brendale, QLD 4500, Australia; 3Brisbane City Council, Program, Planning and Integration, Brisbane Square, Brisbane, QLD 4000, Australia

**Keywords:** *Trichoderma*, streptomycetes, *Pyrrhoderma noxium*, consortium of biological control agents, actinomycetes, wood decay, brown root rot pathogen

## Abstract

The Brown root rot pathogen *Pyrrhoderma noxium* (Corner) L.W. Zhou and Y.C. Dai is known to infect a large number of culturally and economically important plant species across the world. Although chemical control measures have been effective in managing this pathogen, their adverse effects on the ecosystem have limited their use. The use of biological control agents (BCAs) thus is generally accepted as an environmentally friendly way of managing various pathogens. Testing various consortia of the BCAs with different antagonistic mechanisms may even provide better disease protection than the use of a single BCA against aggressive plant pathogens such as the *P. noxium*. In the presented study, the wood decay experiment and the pot trial confirmed that the consortium of *Trichoderma* strains (#5029 and 5001) and streptomycetes (#USC−6914 and #USC−595-B) used was effective in protecting wood decay and plant disease caused by *P. noxium*. Among the treatments, complete elimination of the pathogen was observed when the BCAs were applied as a consortium. In addition, the BCAs used in this study promoted the plant growth. Therefore, *Trichoderma* and streptomycetes consortium could be used as a potential biocontrol measure to manage *P. noxium* infections in the field over the application of hazardous chemical control measures.

## 1. Introduction

The white root fungus, *Pyrrhoderma noxium* (Corner) L.W. Zhou and Y.C. Dai can produce cellulolytic and ligninolytic enzymes, which can play a key role in root decay leading to tree failures and agricultural losses across the tropical and subtropical regions of the globe [1,2,3,4]. Several physical and chemical disease control measures including root barriers, removal of infested plant material, flooding, fumigation and fungicide have been tested against this pathogen by different researchers [1,4,5,6]. Among them, fungicides such as propiconazole, triadimefon, prochloraz and methyl thiocynate were found to be effective against this pathogen [6]. However, due to the detrimental impacts of these fungicides on the environment, humans and animals, the chemical-based control measures are not widely used against *P. noxium*.

In recent years, biological control of phytopathogens by the application of antagonistic microorganisms has gained more attention over the chemical-based control measures [7,8,9,10,11,12]. Strains belonging to the genera of *Trichoderma* and streptomycetes have been widely used in biological control applications and reported as being effective against many phytopathogenic fungi including *Phytophthora capsici, Puccinia triticina, Magnaporthe oryzae*, *Pythium aphanidermatum*, *Phytophthora fragariae, Fusarium verticillioides*, *Rhizoctonia solani*, *Botrytis cinerea*, *P. noxium*, *Pythium ultimum*, *Fusarium oxysporum*, *Fusarium solani*, *Phytophthora nicotianae* and *Sclerotium rolfsii* [4,13,14,15,16,17,18,19,20,21,22,23,24,25]. Furthermore, several green house and field studies have reported that the combined application of fungal and bacterial biocontrol agents with different mechanisms of disease control (competition for nutrients, antibiosis, mycoparasitism, induced resistance and plant growth enhancement) remain the most efficient way of protecting the plants growing in pathogen infested soils compared to the use of a single biocontrol agent [17,19,26,27].

Previously, several *Trichoderma* and streptomycetes were tested as a single biocontrol agent against *P. noxium* in plate assays as well as in in vitro wood decay experiments and green house studies [2,3,4,28,29]. However, there were no prior attempts made on testing a consortium of *Trichoderma* and streptomycetes for the management of this pathogen.

In biological control studies, in vitro evaluation of antagonist against the pathogen on an agar medium provides an initial screening for its antagonistic ability but does not mean that the same effect is reproduced in container media or in soil [30]. Therefore, observed in vitro biological control potential of the antagonist should be re-tested in container media or in soil. This study thus evaluated both in vitro and in vivo antagonistic activities of the BCAs as single biocontrol agents (either the *Trichoderma* strains or the streptomycetes inoculum) and as a combined inoculum against *P. noxium* in soil. Therefore, the aims of this study were (1) to assess the biocontrol efficacy of the BCAs in preventing in vitro wood decay caused by *P. noxium*; (2) to test the BCAs for disease protection of plants growing in *P. noxium* infested soil; and (3) to evaluate the effect of *P. noxium* and the BCAs on plant growth.

## 2. Materials and Methods

### 2.1. Details of the Biological Control Agents and Pathogen Used

Two streptomycetes (#USC−6914: OP592219, closely related to *Streptomyces cinereus* and #USC−595-B: OP592220, closely related to *Streptomyces platensis*) [31] and two *Trichoderma* strains (#5001: SAMN30380231 and #5029: SAMN30380232) closely related to *Trichoderma reesei* [32] were tested as BCA against *P. noxium* consortium (strains A, B, D, E and F).

### 2.2. In Vitro Evaluation of the Antagonistic Potential of Trichoderma Strains and Streptomycetes on P. noxium Using Wood Block Test

The wood decay experiment was conducted using the method described by Schwarze et al. [2] with slight modifications as follows. Freshly cut wood blocks (20 mm × 20 mm × 20 mm) obtained from a *Ficus macrophylla* (located at Pinkenba, Queensland Australia) on 8 April 2020 were dried in an oven (10 April 2020 to 12 April 2020) at 105 °C until a constant weight was obtained. The dry weight of each wood block was recorded before autoclaving twice at 15 lbs of pressure at 121 °C for 15 min using the dry cycle program to prevent the accumulation of moisture. A final autoclave cycle was performed 24 h before the start of the experiment (14 April 2020). All blocks were re-wetted using sterile distilled water for 3 min under reduced pressure in a vacuum desiccator to increase their moisture content to 50% (by dry weight) [33].

Gamma irradiated (for 48 h; 10 April 2020 to 12 April 2020 with 50 kGy) topsoil (200 g, pH 6.5) was packed in sterile polyproplene containers with air filter embedded lids, and the soil moisture level was maintained at 65% WHC by adding sterile distilled water. Two *Trichoderma* strains (#5001 and #5029) and streptomycetes (#USC−6914 and #USC−595-B) previously confirmed for their antagonistic activity against *P. noxium* in vitro were selected as the test organisms [32]. *P. noxium* consortium was prepared by inoculating 6 plugs (0.5 cm radius) of *P. noxium* strains A, B, D, E and F onto sterile millet seeds on 15 March 2020). *Trichoderma* consortium was prepared by the co-inoculation of *Trichoderma* strains #5001 and #5029 (15 plugs/strain used for inoculation), while a streptomycetes consortium was prepared by the co-inoculation of streptomycete #USC−6914 and streptomycete #USC−595−B (15 plugs/species used for inoculation) on 15 March 2020. After visual confirmation of the inoculum establishment on the millet seeds (after 30 days), inoculum (2% *w/w*) was thoroughly mixed with the sterile soil packed in the sterile containers. The inocula combinations used in this experiment are listed in Table 1. 

Due to the slow growth rate, *P. noxium* and streptomycetes were inoculated seven days before wood blocks placed on the packed soil (8 April 2020). On the other hand, *Trichoderma* was inoculated on the start of the experiment date (15 April 2020) due to its fast growth rate. Finally, two sterile wood blocks were aseptically placed into each sterile container and incubated for a period of three months (15 April 2020 to 15 July 2020) at 28 ± 1 °C. The treatments (Table 1) were arranged in randomized block design [4] comprising two blocks with ten replicates for each treatment. A total of twenty wood blocks were used per treatment.

#### 2.2.1. Effect of BCAs in Protecting the Weight Loss in *P. noxium* Treated Wood Blocks

The ability of the BCAs to protect the wood blocks from the pathogen infestation was evaluated in terms of percentage weight loss in the BCA treated wood blocks. Ten wood blocks randomly selected from each treatment (on 15 July 2020) were cleaned and oven-dried at 70 °C until a constant weight was obtained. Final dry weights were recorded and used for calculating the percentage weight loss of each block [33].

#### 2.2.2. Effect of BCAs on Protecting the Strength of *P. noxium* Treated Wood Blocks

The ability of the BCAs to protect the wood strength by controlling the growth of *P. noxium* was studied using a Longitudinal Compression Strength (LCS) test [34]. Five randomly selected wood blocks representing each treatment were cleaned (on 15 July 2020) and placed on a specially designed metal jig of an Instron testing machine (Shimadzu Autograph AG-X plus, 100 kN), where a longitudinal axis of the wood was set parallel to the loading direction. The wood samples were compressed at a crosshead speed of 0.3 mm/min. Compressive strength required to compress 5% of the original dimension of the wood block was recorded. The load required for this compression was used as a measure of residual strength [34]. All the wood blocks had the same surface area exposed to the loading, compressed to the same percentage of their length [34]. LCS values were recorded as newtons (N).

#### 2.2.3. Inhibitory Effect of Biological Control Agents on the Establishment of *P. noxium* on Wood Blocks

The inhibitory effect of the biological control agents on establishment of *P. noxium* on wood blocks was evaluated in terms of the frequency of pathogen re-isolation from inner and outer wood surfaces. Five wood blocks randomly selected from the treatments of P, SP, TP and TSP were cleaned (to remove mycelium and soil particles) inside the biosafety cabinet (on 15 July 2020). A total of 25 wood fragments per treatment comprising five fragments per block were removed from outer surface. For outer surface sampling, one wood fragment was taken from top and four sides of the blocks facing upwards. A specimen was not taken from the bottom surface since it was already in contact with the soil inoculated with pathogen/BCAs. After splitting the wood blocks at the center, 25 wood segments were taken from the center part of blocks for each treatment comprising five wood segments per block. This experiment was performed only once. The wood fragments were inoculated on modified *P. noxium* selective media [35]. The inoculated plates were incubated at 28 °C for 10 days, and the *P. noxium* growth was recorded. Finally, effectiveness of BCAs on controlling *P. noxium* was expressed as percentage of eradicated *P. noxium* [3].

### 2.3. Pot Experiment

A pot experiment was carried out to evaluate the antagonistic abilities of *Trichoderma* strains and streptomycetes singly or combined against *P. noxium* in the greenhouse settings. *F. macrophylla* was selected as the test plant due to its cultural significance in Queensland, Australia. The experiment was conducted in a shade house located at Seventeen Mile Rocks, Brisbane, Australia for a period of six months between 10 December 2018 and 10 June 2019.

#### 2.3.1. Treatment Selection and Container Media Inoculations

Inocula details used in the experiment are listed in Table 1, and millet seed was used as carrier medium. Inoculum (2% *w/w*) was thoroughly mixed with 2.5 kg of gamma irradiated (for 48 h; 7 December 2018 to 9 December 2018 with 50 kGy) potting mix as per treatment requirements at the time of planting. Approximately same sized (15 cm) and aged (six months) *F. macrophylla* plants produced from cuttings were purchased from Ibrox Park Nursery, Burbank, Australia on 10 December 2018 were used in this study. Roots were surface sterilized on 10 December 2018 in 0.5% sodium hypochlorite solution for 2 min to prevent any existing pathogenic fungi from flourishing during the experiment. After the surface sterilization of roots, plants were re-potted in 200 mm plastic pots filled with potting mix and inocula combinations (Table 1). Controls were treated with un-inoculated millet seeds. Enough spacing was provided between the pots to prevent cross contamination. Soil moisture level (WHC, 65%) and temperature (30 ± 2 °C) were maintained throughout the experiment. This experiment was carried out following a completely randomized design with four plants per treatment [36]. This experiment was performed only once.

After two weeks (on 24 December 2018) of inoculum application, re-isolation of *P. noxium* and the BCAs was attempted from the potting mix of each inoculated pot to confirm the establishment of the inocula. One gram of potting mix collected from each treatment was serially diluted up to 10^−7^ and 100 µL of aliquots from this dilution were inoculated onto selective media. For *P. noxium,* modified *P. noxium* (supplemented with 100 mg/l streptomycin sulfate) [35], for streptomycetes, starch casein agar [37] and for *Trichoderma* strains, *Trichoderma* agar media, were used [38,39]. Plates were incubated at 28 °C for *Trichoderma* and *P. noxium* and at 25 °C for streptomycetes for seven days. Triplicate plates were used for each treatment. The *Trichoderma*, streptomycetes and *P. noxium* resembling colonies were isolated from the plates and were identified based on the morphological appearance of the original cultures used in the study.

#### 2.3.2. Effect of Pathogen and Biological Control Agents on the Growth Parameters of Fig Plants

At the end of six months (on 10 June 2019), fig plants were carefully removed from pots, and soil particles attached on the roots were cleaned without damaging the roots. Shoot length was measured between plant collar (root initiation point) and terminal shoot point. Roots were straightened, and the length was measured between root initiation point to the terminal root. After separation, fresh root and shoot weight of each plant were recorded. Shoots and roots were dried at 70 °C for 7 days before obtaining the dry shoot and root weight. Shoot length, root length, fresh shoot weight, dry shoot weight, fresh root weight and dry root weight of inoculated plants were compared with the un-inoculated control plants to study the effect of inocula on the plant growth parameters. Similarly, disease control ability of the BCAs was evaluated by comparing the growth parameters of plants treated with the BCAs and *P. noxium* and the plants treated only with *P. noxium*.

#### 2.3.3. Inhibitory Effect of Biological Control Agents on the Establishment of *P. noxium* in Plant Roots

The ability of the BCAs to control the growth of *P. noxium* and prevent the disease was evaluated in terms of frequency of *P. noxium* re-isolation from the root samples. At the end of the experiment (on 10 June 2019), 15 root cuttings were collected from two main and five fine roots [4] of each fig plants representing the treatments of P, SP, TP and TSP. They were then washed for 5 min under running water, 75% ethanol for 40 seconds, 0.5% sodium hypochlorite solution for 40 seconds, rinsed thrice with sterile distilled water for 15 seconds, and ~1 cm root segments were inoculated on the *P. noxium* selective medium [4]. A total of 60 root segments from each treatment were used in the re-isolation step. Inoculated plates were incubated at 28 °C for 5 days. A biological control effect of each *Trichoderma* strain, streptomycetes and combined BCAs consortia were expressed as a percentage of eradicated *P. noxium* from the roots [4].

### 2.4. Statistical Analysis

For the wood block study, the effect of treatments on the wood weight and strength was evaluated via one-way ANOVA followed by Tukey’s test using SPSS statistics 26 [40]. Similarly, the effect of treatments on the growth parameters of the plants used in the pot experiment was also analysed using one-way ANOVA followed by Tukey’s test using SPSS statistics 26 [40]. 

## 3. Results

### 3.1. In Vitro Evaluation of the Antagonistic Potential of Trichoderma Strains and Streptomycetes on P. noxium Using a Wood Block Test

The wood blocks treated with *P. noxium* showed decayed signs with a conspicuous network of brown lines while the blocks treated with BCAs did not show any sign of *P. noxium* degradation (Figure 1).

#### 3.1.1. Effect of BCAs in Preventing the Weight Loss and Strength of *P. noxium* Treated Wood Blocks

There was significant difference between the treatments in weight loss percentage (*F*_(7,72)_ = 1392.9, *p* = 0.001) and wood strength loss (*F*_(7,32)_ = 49.13, *p* < 0.001). The wood blocks from all the treatment showed some amount of weight and strength loss compared to control (Figure 2). However, the weight and strength loss in the blocks treated with *P. noxium* was significantly (*p* = 0.001) higher than the blocks treated with BCA (T or S or TS). On the other hand, the weight and strength loss in the treatments with BCAs and *P. noxium* (SP or TP or TSP) was significantly lower (*p* = 0.001) than the weight loss measured in *P. noxium* (P) treatment (Figure 2). Weight loss percentage in the treatments with BCAs and *P. noxium* was ~5 times lower than the weight loss percentage measured in treatment with *P. noxium* (Figure 2). Meanwhile, the strength of the blocks measured in the treatments with BCA, and *P. noxium* was approximately ~1.6 times higher than the wood strength measured in *P. noxium* treatment (Figure 2). The weight and strength loss recorded between the treatments of TP, SP and TSP was not significantly different. These results indicate that BCAs were effective as T or S or when used as combined (TS) against *P. noxium* in preventing the wood decay.

#### 3.1.2. Inhibitory Effect of Biological Control Agents on the Establishment of *P. noxium* on Wood Blocks

The re-isolation frequency of *P. noxium* was low in the blocks treated with BCAs and *P. noxium* (TP or SP or TSP) compared to the wood blocks treated with *P. noxium* (Table 2). Complete elimination of *P. noxium* was noticed in the treatment with *P. noxium* and all the BCAs (TSP) compared to *Trichoderma* consortium with *P. noxium* (TP) or streptomycetes consortium with *P. noxium* (SP). Penetration of *P. noxium* into the inner wood surface also prevented in the treatments of TP, SP and TSP, while *P. noxium* was detected in the inner wood surface in the treatment with *P. noxium* (P) (Table 2). 

### 3.2. Pot Experiment

#### 3.2.1. Re-Isolation of the Pathogen and BCA from Container Media

Initial screening (after 2 weeks of inoculation) confirmed the establishment of the BCAs and *P. noxium* in the potting mix (Figure 3). During the experiment period, none of the plants died due to *P. noxium* infection.

#### 3.2.2. Effect of Pathogen and Biological Control Agents on the Growth Parameters of Fig Plants

There was a significant variation among the treatments in terms of root length (*F*_(7,24)_ = 65.938, *p* < 0.001), shoot length (*F*_(7,24)_ = 169.5, *p* < 0.001), fresh root weight (*F*_(7,24)_ = 191.53, *p* < 0.001), dry root weight (*F*_(7,24)_ = 93.89, *p* < 0.001), fresh shoot weight (*F*_(7,24)_ = 364.326, *p* < 0.001) and dry shoot weight (*F*_(7,24)_ = 157.93, *p* < 0.001) of plants. The root length, shoot length, fresh root weight, dry root weight, fresh shoot weight and dry shoot weight of the plants treated with *P. noxium* were significantly (*p* < 0.001) lower than the controls and the plants grown in the presence of BCAs (T or S or TS or TP or SP or TSP) (Figure 4). On the other hand, root length, shoot length, fresh root weight, dry root weight, fresh shoot weight and dry shoot weight of plants treated with *Trichoderma* consortium (T), streptomycetes consortium (S), streptomycetes with *Trichoderma* consortium (TS), *Trichoderma* consortium with *P. noxium* (TP), streptomycetes consortium with *P. noxium* (SP) and *P. noxium* with all the BCAs (TSP) were significantly (*p* < 0.001) higher than control plants (Figure 4). In the absence of *P. noxium,* the root length, shoot length, fresh root weight, dry root weight, fresh shoot weight and dry shoot weight in the plants inoculated with the combined BCAs (TS) was significantly (*p* < 0.001) higher than the plants inoculated with *Trichoderma* consortium or streptomycetes consortium. Similarly, in the presence of *P. noxium*, shoot length and root length of plants treated with the combined BCAs (TSP) were significantly (*p* < 0.001) higher than the plants treated with *Trichoderma* or streptomycetes consortium (Figure 4).

#### 3.2.3. Inhibitory Effect of Biological Control Agents on the Establishment of *P. noxium* in Plant Roots

Isolation frequency of *P. noxium* from the plants grown in the *P. noxium* treated soil was higher than all the other treatments (TP, SP and TSP) (Table 3). The combined BCAs consortium treatment completely inhibited the pathogen establishment on plant roots compared to the treatments of *Trichoderma* consortium with *P. noxium* (TP) and streptomycetes consortium with *P. noxium* (AP). Furthermore, *P. noxium* re-isolation frequency was slightly higher in treatment with streptomycetes consortium compared to the treatment with *Trichoderma* consortium.

## 4. Discussion

The presented study evaluated the impact of *P. noxium* on wood and plants using laboratory and green house assays. *P. noxium* treated wood blocks in the absence of BCAs showed ~23% of weight loss and possibly cell wall degrading enzymes produced by *P. noxium* played an important role in the wood degradation [41]. Similar findings were previously reported by Schwarze el al. [2], Chou et al. [4] and Burcham et al. [42], and the weight loss caused by the *P. noxium* ranged from 0.94 to 34.78%, 3.5 to 11.5%, 12.7 and 10.8%, respectively. The variation in the weight loss percentage could be due to the *P. noxium* strains and the wood species used in the experiments. The wood degradation also correlates with the decline of properties such as tension or bending [43,44]. This could be the reason in this study that a significant reduction of wood strength was observed in the wood blocks treated with *P. noxium* (P) compared to other treatments.

The lowest re-isolation percentage of the pathogen in the presence of *Trichoderma* strains (#5001 and #5029) and streptomycetes (#USC−6914 and #USC−595−B) indicates that the BCAs used in this study were effective in preventing the establishment of pathogen on the blocks, thus protecting the wood blocks from weight and strength loss. Similarly, *T. virens*, *T. asperellum*. *T. koningiopsis*, *T. harzianum*, *T. reesei* and *T. saturniporum* used in previous studies were also reported as effective in controlling *P. noxium* growth and protecting wood blocks [2,3,4,42]. In a different study, *Streptomyces atratus* and *S. tsukiyonensis* were effective against brown rot fungus, *Gloeophyllum trabeum*, and protected the wood properties [43]. When the *Trichoderma* strains (#5001 and #5029) and streptomycetes (#USC-6914 and #USC-595-B) were used together, possibly the combined biological control mechanisms [45,46,47,48,49,50,51,52] of them resulted in the induced antagonistic activity against the pathogen and provided better protection than the single BCA application. Therefore, these BCAs can be used together as an effective management tool to protect wood decay from *P. noxium*. In addition, these control agents may be used as environmentally friendly wood preservatives over toxic chemicals used in the protection of construction materials.

In this study, the growth parameters of the *P. noxium* (P) treated plants were significantly reduced compared to other treatments. Similar negative impact by *Phythium ultimum*, *P. aphanidermatum, Phytopthora capsici*, *F. solani*, *R. solani*, on plant growth parameters of different plant varieties were previously reported in a number of greenhouse experiments [4,15,18,22,25,53]. *Trichoderma* strains (#5001 and #5029) used in this study were effective in protecting the plants from *P. noxium* infections. Similar results were also previously reported with antagonistic *T. asperellum* against *P. noxium* [4], with *T. longibrachiatum* against *Fusarium solani* [18] and with *T. viride* against *R. solani* [15] in greenhouse and field experiments. As with *Trichoderma* inoculants, streptomycetes (#USC-6914 and #USC-595-B) used in this study were also found to be effective in protecting the plants from *P. noxium* infection. Similarly, antagonistic ability of *S. tsusimaensis*, *S. caviscabies*, *S. setonii*, *S. goshikiensis*, *S. africanus* against *Fusarium oxysporum* [54], *S. philanthi* against *Rhizoctonia solani*, *S. sindeneusis* isolate #263 and *S. globisporus* JK-1 against *Magnaporthe oryzae* [55] and *S. cinereus* against *F. oxysporum*, *R. solani*, *Aspergillus niger*, *Alternaria brassicicola* and *Phytophthora dresclea* [56] have also been previously reported.

Since the *Trichoderma* and streptomycetes have different modes of antagonistic mechanisms against plant pathogens and various benefits for the plant growth and development, a number of studies have focused on developing a consortium of *Trichoderma* and streptomycetes for biological control applications [23,26]. Microbial co-cultivation has also been reported to induce the expression of several cryptic pathways and increase the production of new secondary metabolites through the activation of signaling molecules [23,57,58]. This could be the reason why *Trichoderma* strains (#5001 and #5029) and streptomycetes (#USC-6914 and #USC-595-B) were inoculated together, and enhanced antagonistic activity was displayed against *P. noxium*, which resulted in complete inhibition of *P. noxium* growth. Similarly, a consortium of *S. viridosporus* and *T. harzianum* and consortium of *T. harzianum* and *S. rochei* used in previous studies showed an enhanced inhibitory effect against *Puccinia triticina* and *Phytophthora capsici*, respectively, compared to a single strain application [17,19]. However, previously the combined inoculum application had not been tested against *P. noxium*. The presented study clearly showed that consortium of *Trichoderma* strains (#5001 and #5029) with the streptomycetes (#USC-6914 and #USC-595-B) may be considered as a potential biological control measure to manage the pathogen and protect the plants growing in the *P. noxium* infested soils. In addition, application of consortia inoculum can also provide a greater protection against genetically diverse *P. noxium* strains in the field. Further studies are underway to evaluate their biological control efficacy in the field.

The BCAs used in this study were also found to promote the plant growth. The plant growth promotion by *Trichoderma* and streptomycetes is due to the production of plant regulators or production of secondary metabolites which might be associated with solubilisation and mobilisation of micronutrients and mineral ions [59,60,61,62,63]. A possible combination of such reported mechanisms might be the reason for the growth promotion observed in the T, TP, S, SP and TSP treatments. Further studies are underway to evaluate the plant growth promoting ability of BCAs in the field.

## 5. Conclusions

The wood decay study and pot experiment results indicate that the application of a consortium of *Trichoderma* strains and streptomycetes can be used as an effective way of controlling *P. noxium.* This can be more effective than the use of individual biocontrol agents in the Brisbane Hinterland and other geographical settings to control similar infections. In addition, the plant growth promoting ability of the BCAs observed in this study also suggests that these BCAs can facilitate the recovery of *P. noxium* infected plants in soil.

## Figures and Tables

**Figure 1 jof-09-00067-f001:**
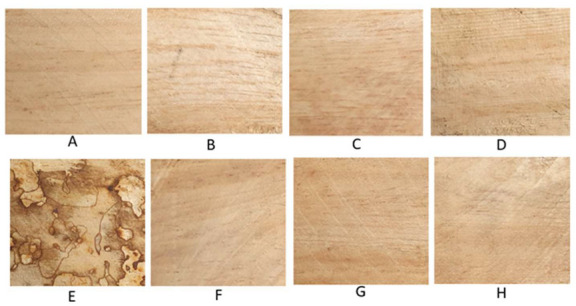
Appearances of cleaned and oven dried wood blocks subjected to various treatments (at the completion of the experiment), (**A**) Control (C); (**B**) *Trichoderma* consortium (T); (**C**) streptomycetes consortium (S); (**D**) *Trichoderma* and streptomycetes consortium (TS); (**E**) *P. noxium* (P); (**F**) *Trichoderma* consortium and *P. noxium* (TP); (**G**) streptomycetes consortium and *P. noxium* (SP); (**H**) *P. noxium* with all the BCAs (TSP).

**Figure 2 jof-09-00067-f002:**
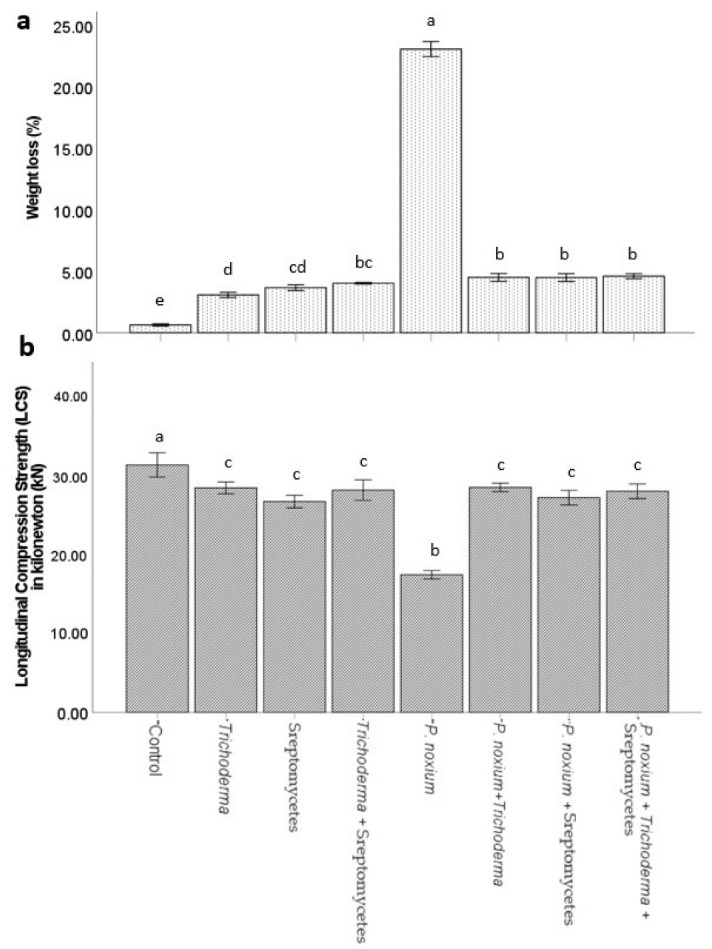
Inhibitory effects of biological control agents on *P. noxium* and subsequent prevention of wood decay (**a**) weight loss (%) and (**b**) wood strength. Footnote: Error bars represent standard error of the mean, and statistically significant (*p* < 0.05) differences are noted by different letters (above the bars) according to Tukey’s test.

**Figure 3 jof-09-00067-f003:**
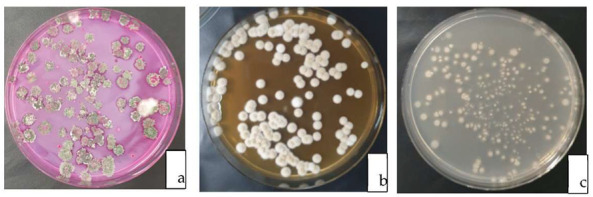
Re-isolation of (**a**) *Trichoderma*, (**b**) *P. noxium* and (**c**) streptomycetes on their respective selective media after two weeks of inoculation into the container media.

**Figure 4 jof-09-00067-f004:**
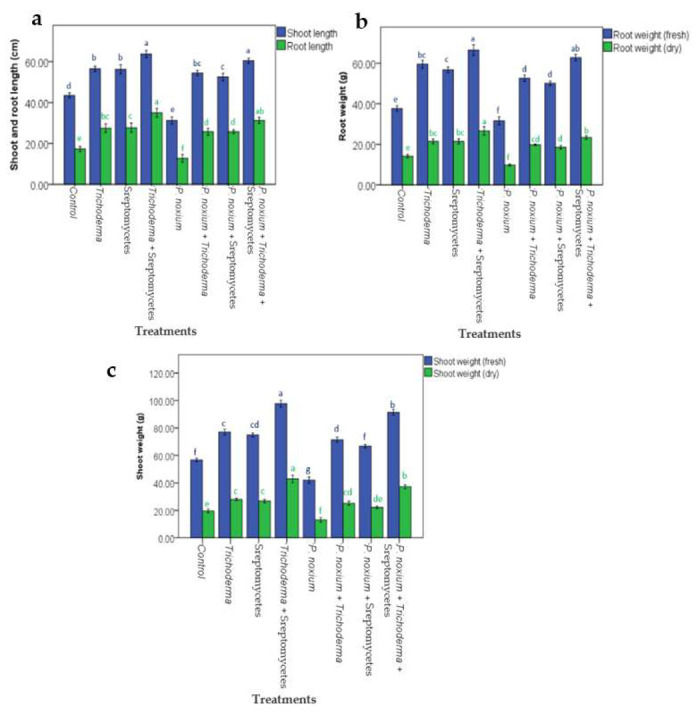
Effect of different treatments on (**a**) plant shoot and root length; (**b**) fresh and dry root weight and (**c**) fresh and dry shoot weight. Footnote: Different letters with the same color on the top of SE bars indicate significant differences in a particular growth parameter between the treatments based on Tukey’s test, *p* < 0.05.

**Table 1 jof-09-00067-t001:** Detail of the inocula used in each treatment.

Treatment	Inocula
*Trichoderma* consortium (T)	*Trichoderma* strains (#5001 and #5029)
Streptomycetes consortium (S)	Streptomycetes (#USC−6914 and #USC−595−B)
*P. noxium* (P)	*P. noxium* strains (A, B, D, E and F)
*Trichoderma* consortium and streptomycetes consortium together (TS)	*Trichoderma* strains (#5001 and #5029) with streptomycetes (#USC−6914 and #USC−595−B)
Streptomycetes consortium with *P. noxium* (SP)	*P. noxium* strains (A, B, D, E and F) with streptomycetes (#USC-6914 and #USC−595−B)
*Trichoderma* consortium with *P. noxium* (TP)	*P. noxium* strains (A, B, D, E and F) with *Trichoderma* strains (#5001 and #5029)
*Trichoderma* consortium, streptomycetes consortium and *P. noxium* (TSP)	*P. noxium* strains (A, B, D, E and F) with *Trichoderma* strains (#5001 and #5029) and streptomycetes (#USC−6914 and #USC−595−B)
Control	Without any inoculation of the above listed BCAs or the pathogen

**Table 2 jof-09-00067-t002:** Re-isolation frequency of *P. noxium* from the outer and inner surface of wood blocks treated with *P. noxium*.

Treatments	Ratio of Outer Wood Segments *	Isolation Frequency (Outer Surface) %	Ratio of Inner Wood Segments *	Isolation Frequency (Inner Surface) %
*P. noxium* (P)	23/25	92	25/25	100
*Trichoderma* consortium with *P. noxium* (TP)	2/25	8	0/25	0
Streptomycetes consortium with *P. noxium* (SP)	3/25	12	0/25	0
All the BCAs with *P. noxium* (TSP)	0/25	0	0/25	0

* Number of wood specimens from *P. noxium* re-isolated/total number of wood specimens.

**Table 3 jof-09-00067-t003:** Re-isolation frequency of *P. noxium* from the root samples of the Fig plants.

Treatments	Ratio of Root Fragments *	*P. noxium* Isolation Frequency (%)
*P. noxium* (P)	52/60	86.67
*Trichoderma* consortium + *P. noxium* (TP)	5/60	8.33
Streptomycetes consortium + *P. noxium* (SP)	7/60	11.67
*P. noxium* with all the BCAs (TSP)	0/60	0

* Number of root fragments from *P. noxium* re-isolated/total number of root fragments.

## Data Availability

Genome data of *Trichoderma* strain #5001 and #5029 are deposited in NCBI under accession numbers of SAMN30380231 and SAMN30380232, respectively. ITS sequences of *P. noxium* strain A, B, D, E and F are deposited under GenBank accession numbers of OP430849, OP430850, OP430851, OP430852 and OP430853, respectively. Similarly, 16srRNA sequences of streptomycetes USC-595-B and USC-6914 are also deposited in NCBI under the accession numbers of OP592220 and OP592219, respectively.

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
