# Peer review of "Testing the Biocontrol Ability of a Trichoderma-Streptomycetes Consortium against Pyrrhoderma noxium (Corner) L.W. Zhou and Y.C. Dai in Soil"

_jof, 2022, doi:10.3390/jof9010067_

Round 1

Reviewer 1 Report

Please see the pdf file.

Author Response

Reviewer 1

  1. The authors report the suppression of the pathogen growth, Pyrrhoderma noxium, by the addition of strains of two genera, Trichoderma and Streptomyces. There can be one problem in their study. I was not able to evaluate this study in reading the manuscript due to sufficient description on the methods whether the study was planned in rationale and logical designs or not. They provide the data of the efficacy of microbiocontrol agents on the growth of fig plants, showing Fig. 4 on page 5. They mention how they evaluated the efficacy in L162-167. However, they do not describe how they prepared the materials. This makes it impossible to determine the validity of the experiment. They have to state how they sampled fig plants, how they assigned their treatments to these trees and how they collected the data from the materials. Ideally, one treatment should have been made on one plant. Of course, other possible assignments could have been considered. However, if they wrongly sampled the trees to which assign treatments, their results would make no sense in statistical meaning.

Additional details provided in the method section

            Other minor comments are written below.

  1. Date and place of material preparation

The authors have to describe individually for the date when they collected the materials and made any treatments. Also the place(s) where the authors obtained the materials should be provided.

Information on the experimental dates and the place where materials obtained from are included

  1. L13

Refer to the author(s) of the species. All species which were used for this study must be followed by the authors when they mention first in the manuscript.

Abbreviations are corrected and in addition, at the genus level Streptomyces was used, whereas at the species level plural form of streptomycetes and singular form of streptomycete are used.

  1. L23 and L24

What species of these genera were used in this study? They are not referred to throughout this manuscript.

Accession numbers for these species are provided. closest relatives of Streptomyces spp were Streptomyces platensis (#USC−595-B) and Streptomyces cinereus (#USC−6914), while Trichoderma reesei was closest relative of Trichoderma strains#5001 and #5029.

Spell out the genera when they are referred to first in the text.

Pathogen names are fully spelt out in the text

  1. L75 modification

Explain briefly what procedures in the method of Schwarze et al were modified how.

Modified steps used in this study is explained in L78 to L104

  1. L83

Refer to the duration of the irradiation.

Reported in L 86, 48 hours of irradiation at 50kGy was used

  1. L85-87

Citation is needed for the confirmation.

References are included

  1. L103

Italicise the species name. This should be applied all, not individually pointed out here.

Throughout the paper scientific names are italicised

  1. L141

Refer to the origin of the plants and also the conditions under which they had been maintained until the experiment.

Origin of plants is included in L145. These plants were purchased from the nursery on the starting day of the experiment. Therefore, plants were not maintained at the experiment site. However, the conditions which we provide during the experiment period is mentioned in L151 and L152

  1. L153

Mention the dilution interval.

Soil sampling was done after two weeks of inoculum application and sample were serially diluted up to 10-7 and tested for the establishment of the inoculum.

  1. L155

Explain briefly the modification.

100 mg/l streptomycin sulfate was used instead of 100 mg/l ampicillin

  1. L158

In this manuscript, streptomycetes and Streptomyces both are used. If they are the same, use only one term throughout the manuscript in order to avoid confusion.

Throughout the manuscript plural form of streptomycetes and singular form of streptomycete are used.

  1. L163

Explain how the root length was measured. It should have been very tough tasks.

Explained in L168, roots were straightened before taking the root length from the root initiation point to root terminal.

  1. L201

The probability should be “< 0.0001”. Theoretically the probability cannot be equal to “0”, unless the sample number would be infinitive.

Probability is corrected to <0.001

  1. L202

The sentence does not make sense. It should be corrected.

Sentence is corrected as “The wood blocks from all the treatment showed some amount of weight and loss strength loss compared to control (Figure 2).”

  1. L206

What statistical method(s) was (w ere) used? Give the method and statistics with df(s).

Detail of statistical method is provided in L187 to L191 (statistical analysis section). Degrees of freedom reported in results sections.

  1. 2

The data shown for P. noxium seem to be different from the description in the text. It is mentioned in the text that the treatment with Trichoderma was significantly effective to reduce the weight loss (L203-204). However, the weight loss in this treatment was significantly larger than any other treatments, even that with Pnoxium only.

Initially Figure 2a had different order of treatments in the x-axis, when the two graphs were combined into one figure, figure 2a’s x -axis was accidentally removed and caused the contradicted description to the text. X-axis of both graphs in Figure 2 were rearranged.

  1. Also, the order of the treatment on the abscissa should be the same as in Fig. 4.

Treatment order on the abscissa of Figure 2 was rearranged to have same treatment order as Figure 4.

  1. L289-290

Refer to the number of species for each genus used in this study.

The sentence is re arranged as the lowest re-isolation percentage of the pathogen in the presence of Trichoderma strains (#5001 and #5029) and Streptomyces species (#USC−6914 and #USC−595−B).

  1. L332-335

The statement is too early. Since this study was carried out for plants in laboratory and those on pots, but not for field. The use of these species may be considered for but cannot be concluded as the biocontrol agents in field until they are examined in field experiments. One more possibility is that they may really help the plant growth but it “must” be evaluated their efficacy on fruit yields. Plants may often grow faster under some conditions at the mercy of their nutritional investment into reproductive organs. This point should be discussed.

Sentence is re structured as follow “The presented study indicates that consortium of Trichoderma strains (#5001 and #5029) with the Streptomyces species (#USC-6914 and #USC-595-B) may be considered as potential biological control measure to manage the pathogen and protect the plants growing in the P. noxium infested soils”.

The ongoing field study will be used for further evaluating the biological control and plant growth promoting ability of Trichoderma strains (#5001 and #5029) and Streptomyces species (#USC-6914 and #USC-595-B) in the field.

Reviewer 2 Report

The authors report the suppression of the pathogen growth, Pyrrhoderma noxium, by the addition of strains of two genera, Trichoderma and Streptomyces. There can be one problem in their study. I was not able to evaluate this study in reading the manuscript due to sufficient description on the methods whether the study was planned in rationale and logical designs or not. They provide the data of the efficacy of microbiocontrol agents on the growth of fig plants, showing Fig. 4 on page 5. They mention how they evaluated the efficacy in L162-167. However, they do not describe how they prepared the materials. This makes it impossible to determine the validity of the experiment. They have to state how they sampled fig plants, how they assigned their treatments to these trees and how they collected the data from the materials. Ideally, one treatment should have been made on one plant. Of course, other possible assignments could have been considered. However, if they wrongly sampled the trees to which assign treatments, their results would make no sense in statistical meaning.

Other minor comments are written below.

1. Date and place of material preparation

The authors have to describe individually for the date when they collected the materials and made any treatments. Also the place(s) where the authors obtained the materials should be provided.

L13

Refer to the author(s) of the species. All species which were used for this study must be followed by the authors when they mention first in the manuscript.

L23 and L24

What species of these genera were used in this study? They are not referred to throughout this manuscript.

L43-44

Spell out the genera when they are referred to first in the text.

L75 modification

Explain briefly what procedures in the method of Schwarze et al were modified how.

L83

Refer to the duration of the irradiation.

L85-87

Citation is needed for the confirmation.

L103

Italicise the species name. This should be applied all, not individually pointed out here.

L141

Refer to the origin of the plants and also the conditions under which they had been maintained until the experiment.

L153

Mention the dilution interval.

L155

Explain briefly the modification.

L158

In this manuscript, streptomycetes and Streptomyces both are used. If they are the same, use only one term throughout the manuscript in order to avoid confusion.

L163

Explain how the root length was measured. It should have been very tough tasks.

L201

The probability should be “< 0.0001”. Theoretically the probability cannot be equal to “0”, unless the sample number would be infinitive.

L202

The sentence does not make sense. It should be corrected.

L206

What statistical method(s) was (w ere) used? Give the method and statistics with df(s).

Fig. 2

The data shown for P. noxium seem to be different from the description in the text. It is mentioned in the text that the treatment with Trichoderma was significantly effective to reduce the weight loss (L203-204). However, the weight loss in this treatment was significantly larger than any other treatments, even that with P. noxium only.

Also, the order of the treatment on the abscissa should be the same as in Fig. 4.

L289-290

Refer to the number of species for each genus used in this study.

L332-335

The statement is too early. Since this study was carried out for plants in laboratory and those on pots, but not for field. The use of these species may be considered for but cannot be concluded as the biocontrol agents in field until they are examined in field experiments. One more possibility is that they may really help the plant growth but it “must” be evaluated their efficacy on fruit yields. Plants may often grow faster under some conditions at the mercy of their nutritional investment into reproductive organs. This point should be discussed.

Author Response

Reviewer 2

Major comments

  1. The title does not describe the contents of the manuscript. First, this manuscript is not about testing the survival of BCAs. It is about testing the antagonistic effect of BCAs against noxium. Second, the biocontrol efficacy of the BCAs in preventing wood decay and promoting the growth of fig plants was performed in potting soil rather than in container media.

Title is changed as “Testing biocontrol ability of a Trichoderma-Streptomyces consortium against Pyrrhoderma noxium in potting soil”

  1. The results of Figure 2a contradict the description in Line 203-207 of section 3.1.1. The results of Figure 2a clearly showed that blocks treated with S had significantly higher weight loss (approximately 4%) than that treated with noxium (approximately 3%). It is even more surprising that blocks treated with T had the highest weight loss of approximately 23%. Don’t these results indicate that T and S cause weight loss of wood blocks rather than protect wood blocks from weight loss caused by P. noxium? Besides, the blocks treated with BCAs and P. noxium (TP or SP or TSP) had higher weight loss (approximately 5%) than that treated with P. noxium (approximately 3%). These results clearly indicated that BCAs did not protect wood blocks from weight loss caused by P. noxium. Alternatively, the method used for the experiment is incorrect.

Initially figure 2a had different order of treatments in the x-axis, when the two graphs were combined into one figure, figure 2a’s x -axis was accidentally removed and caused the contradicted description to the text. X-axis of both graphs in figure 2 were rearranged and new graphs presented.

  1. In line 98-101, two sterile wood blocks were placed into each container and incubated for three months, and ten replicates for each treatment. However, in line 105, ten wood blocks from each treatment were cleaned for the experiments of Figure 2. I wonder how many wood blocks were used for each treatment. Were the means of the data set from two blocks with 10 replicates or 10 blocks with no replicate?

Ten containers comprising two blocks/container were used per treatment. Therefore, total of twenty blocks represented each treatment. We considered number of containers (10) as the replicate of the treatment. To avoid confusion, the sentence is re written as two blocks with 10 replicates.

  1. The authors need to clarify the following issues concerning Table 2. - It is hard to believe that no pathogen was re-isolated from the center part of any wood block treated with either TP, SP or TSP. - Why did the authors randomly select 5 wood blocks rather than analyze all the 10 blocks used in each treatment as described in line 105? - It is unclear how many wood segments were removed from the outer and inner parts of each wood block, and how many wood segments were removed from each side of the wood block, since the wood were cut into 20 mm x 20 mm x 20 mm cubic blocks. - How many times did the authors replicate the experiments of Table 2?

It was not possible to use all the blocks for re-isolation of pathogen since the wood segments could not be removed from the blocks selected for the strength test. Removal of wood segments would have reduced the strength and would give wrong information about the effect of P. noxium on the reduction of wood strength and also can’t evaluate the ability of BCA in protecting the wood properties. Similarly, five randomly selected blocks for the P. noxium re-isolation were cleaned and all the sampling were done in the biosafety cabinet to prevent the contamination and these blocks can’t be oven dried (it will kill the inoculum).

  1. The data of Figure 4 is uninformative. It is widely acknowledged that beneficial microbes such as Trichoderma and Streptomyces can promote plant growth. It will be more meaningful to assess the biocontrol efficacy of the BCAs on noxium infected plants. In addition, it is unclear how many plants were used and how old these plants were for evaluating the effects of BCAs and pathogen on plant growth in Figure 4. Also, how many times did the authors replicate the experiments of Figure 4?

The biocontrol ability of the BCA was also evaluated in this study using the re-isolation frequency of the P. noxium, please see table 3. In addition to that effect of inocula on plant growth was studied.

The plants used in this study were 6 months old, it is mention in line 145. Four plants/ treatments were used and this is mentioned in line 153. This experiment was only performed once.

  1. For the experiments in Table 3, it is unclear how many plants were used for each treatment, how many roots for each treatment were collected, and how many root segments were collected from each root or each plant and how they were collected.

Four plants were used in each treatment and total of 15 root fragments from two main and five fine roots were collected from each plant to generate 60 root fragments per treatment. Mentioned in line 179

  1. In line 348-350 of Conclusion section, the authors concluded that “the plant growth promoting ability of the BCAs observed in this study suggests that these BCAs can also help the recovery of noxium infected plants in soil”. This conclusion is incorrect, because the authors did not perform any experiment to eliminate P. noxium from P. noxium-infected plants.

This conclusion was made based on the 0% re-isolation frequency observed in the treatment of P. noxium with all the BCAs (TSP) in the pot and wood decay experiments. This indicated that when the Trichoderma and Streptomyces used together as BCAs, P. noxium growth can be completely inhibited, therefore we concluded that if these BCAs used in the field they can protect the plants from P. noxium and assist the recovery of infected plants. Currently field studies are underway to confirm this and the results will be published.

  1. Several sentences were not written correctly. The authors need to check the whole manuscript.

Proof reading is done

Reviewer 3 Report

Congratulations on your interesting work, I have only detected a few small details to correct

Line 87 “Two Trichoderma strains (#5001 and #5029) 85 and two Streptomyces species (#USC−6914 and #USC−595-B) previously confirmed for their 86 antagonistic activity against P. noxium in vitro were selected as the test organisms”

 If it is possible you should include a reference for this previous study.

It is observed in the titles of the different sections the scientific name does not appear in italics:

Line 103 P. noxium-- P. noxium on italic

Line 109 P. noxium-- P. noxium on italic

Line 120 P. noxium-- P. noxium on italic

Line 168 P. noxium-- P. noxium on italic

Line 198 P. noxium-- P. noxium on italic

Line 220 P. noxium-- P. noxium on italic

Line 266 P. noxium-- P. noxium on italic

Author Response

Reviewer 3

Congratulations on your interesting work, I have only detected a few small details to correct

Line 87 “Two Trichoderma strains (#5001 and #5029) 85 and two Streptomyces species (#USC−6914 and #USC−595-B) previously confirmed for their 86 antagonistic activity against P. noxium in vitro were selected as the test organisms”

  1. If it is possible you should include a reference for this previous study.

References are included

It is observed in the titles of the different sections the scientific name does not appear in italics:

  1. Line 103 P. noxium-- noxium on italic

Italics is used in scientific name

  1. Line 109 P. noxium-- noxium on italic

Italics is used in scientific name

  1. Line 120 P. noxium-- noxium on italic

Italics is used in scien ific name

  1. Line 168 P. noxium-- noxium on italic

Italics is used in scientific name

  1. Line 198 P. noxium-- noxium on italic

Italics is used in scientific name

  1. Line 220 P. noxium-- noxium on italic

Italics is used in scientific name

  1. Line 266 P. noxium-- noxium on italic

Italics is used in scientific name

Round 2

Reviewer 1 Report

No further comments 

Author Response

No specific comments are made by the reviewer, method details are added to the text as per Reviewer 2’s comments. English was checked again by a native speaker.

Reviewer 2 Report

The manuscript can be published, if the following three points are solved:

1) The author(s) of species

Any species that were used in or for the study should be followed by their author(s).

2) Date of treatments

The date (day, month, year) of treatments should be referred to. For example, when were tree blocks dried (L84), when was the soil irradiated (L90), and so on.

3) Not recommended to use yet

The first two sentences in Conclusion are only suggestions (not confirmed). This means that the used of the microorganisms used in this study in nature is still in debate. Thus, "can be used " (L449) should be changed for "may be used" or "can be considered for". The last sentence must be re-considered.

Author Response

1) The author(s) of species

Any species that were used in or for the study should be followed by their author(s).

First descriptors names are added for the fungal species. It is not the usual practice for actinomycete species. Rules specific to Mycology and Bacteriology disciplines are followed.

2) Date of treatments

The date (day, month, year) of treatments should be referred to. For example, when were tree blocks dried (L84), when was the soil irradiated (L90), and so on.

Experiment dates are included.

3) Not recommended to use yet

The first two sentences in Conclusion are only suggestions (not confirmed). This means that the used of the microorganisms used in this study in nature is still in debate. Thus, "can be used " (L449) should be changed for "may be used" or "can be considered for". The last sentence must be re-considered.

Conclusion is rewritten as follows”:

The wood decay study and pot experiment results indicate that the application of a consortium of Trichoderma strains and streptomycetes can be used as an effective way of controlling P. noxium. This can be more effective than the use of individual biocontrol agents in the Brisbane Hinterland and other geographical settings to control similar infections. In addition, the plant growth promoting ability of the BCAs observed in this study also suggests that these BCAs can facilitate recovery of P. noxium infected plants in soil.